# Research

evolution

sexual conflict, kin selection, mate harm, *Callosobruchus maculatus*

**Author for correspondence:**
Elena C. Berg
e-mail: eberg@aup.edu

# Kin but less than kind: within-group male relatedness does not increase female fitness in seed beetles

Elena C. Berg[1], Martin I. Lind[2], Shannon Monahan[1], Sophie Bricout[1] and Alexei A. Maklakov[2,3]

[1]Department of Computer Science, Mathematics, and Environmental Science, The American University of Paris, Paris, France
[2]Department of Ecology and Genetics, Animal Ecology, Uppsala University, Uppsala, Sweden
[3]School of Biological Sciences, University of East Anglia, Norwich Research Park, Norwich, UK

ECB, 0000-0002-9637-8109; MIL, 0000-0001-5602-1933; AAM, 0000-0002-5809-1203

Theory maintains within-group male relatedness can mediate sexual conflict by reducing male–male competition and collateral harm to females. We tested whether male relatedness can lessen female harm in the seed beetle *Callosobruchus maculatus*. Male relatedness did not influence female lifetime reproductive success or individual fitness across two different ecologically relevant scenarios of mating competition. However, male relatedness marginally improved female survival. Because male relatedness improved female survival in late life when *C. maculatus* females are no longer producing offspring, our results do not provide support for the role of within-group male relatedness in mediating sexual conflict. The fact that male relatedness improves the post-reproductive part of the female life cycle strongly suggests that the effect is non-adaptive. We discuss adaptive and non-adaptive mechanisms that could result in reduced female harm in this and previous studies, and suggest that cognitive error is a likely explanation.

## 1. Introduction

Males and females have different routes to successful reproduction [1], and this can lead to evolutionary conflict between the sexes [2–4]. One extreme form of this conflict is mate harm, when one sex (usually the male) physically injures the opposite sex (usually the female) [4,5]. Male mate harm occurs in many animals and has been especially well studied in insects, including the bed bug *Cimex lectularius*, the seed beetle *Callosobruchus maculatus* and the fruit fly *Drosophila melanogaster*. Male bed bugs stab females with their genitalia, inseminating females directly into the abdominal cavity [6,7]. Male seed beetles have spines on their intromittent organs that pierce holes in the female's genital tract, reducing female longevity [8–10]. In the fruit fly, males may harass females during courtship [11–14] or physically harm them during the mating process [11,15]. *Drosophila* and *C. maculatus* ejaculates also contain accessory gland proteins that modulate female reproductive behaviour [5,16–18]. Mate harm can evolve for two reasons. One possibility is that the harm itself increases male fitness by causing females to allocate more resources into current reproduction and away from future reproductive attempts with other males [19]. A better-supported alternative in most cases is that mate harm is simply a deleterious side effect of male–male competition over fertilization [4,20,21].

It has been suggested that within-group male relatedness or familiarity can reduce some of the negative effects of sexual conflict by reducing male-induced harm to females. For example, kin selection theory [22,23] suggests that the level of genetic relatedness among competing males can moderate sexual conflict in viscous populations (i.e. populations with genetic structure) [24–27]. In

populations in which close adult relatives are likely to interact, males would gain indirect fitness benefits by helping their close male kin to reproduce. Such cooperation might take the form of reducing mate harm to facilitate sexual access to those females. As Chippindale *et al.* [28] point out, for this kind of kin selection to occur, three conditions must be met: (i) males must harm their mates in some way, thereby reducing female reproductive success; (ii) there must be some mechanism in place for reliably recognizing kin; and (iii) groups of related males must have a reasonable chance of encountering each other during the reproductive period. Another possibility for reduced harm to females is familiarity among males raised together that can lead to reduced levels of aggression [29]. This explanation may be particularly suitable to the systems in which animals spend considerable time together during development. Finally, it is possible that this effect is non-adaptive and results from perception error because of increased phenotypic similarity among individuals kept in a homogeneous environment [29].

A series of recent studies on *D. melanogaster* has investigated this possible role of kin selection in moderating mate harm, with conflicting results [28–33]. In support of the kin selection hypothesis, Carazo *et al.* [30] found that females exposed to groups of three full-sib brothers did indeed have higher lifetime reproductive success and slower reproductive ageing than females exposed to trios of unrelated males. One important point to consider in the Carazo *et al.* [30] study is that it confounded familiarity and relatedness. Brothers used in the study had also been reared together in the same vial, whereas the unrelated males had been raised in separate vials. Hollis *et al.* [29] conducted a follow-up study on a different population of *Drosophila* in which they controlled for familiarity by testing the effect of brothers raised together versus apart. Females exposed to brothers raised together had higher lifetime reproductive success, but this effect disappeared when females were exposed to brothers raised apart. Hollis *et al.* [29] concluded that familiarity and not relatedness *per se* was likely driving the patterns Carazo *et al.* [30] observed.

Chippindale *et al.* [28] performed a fully crossed experiment in which they exposed females to brothers that had been raised together, brothers that had been raised apart and unrelated males that had been raised apart. Unlike Carazo *et al.* [30], they found no evidence that either familiarity or relatedness among males had any effect on female lifespan or reproductive success, a result corroborated in a separate study by Martin & Long [31]. Finally, a later study by Le Page *et al.* [32] suggested that when males are both related and familiar to each other, they cause reduced female harm in *D. melanogaster*.

These conflicting results suggest that it remains unclear what role within-group male relatedness plays in mediating male–male cooperation and mate harm. Moreover, if we are to understand whether inclusive fitness benefits mediate sexual conflict in the animal kingdom, we need to expand our research focus into other model systems. One excellent candidate is the seed beetle *C. maculatus*. As described above, male seed beetles inflict physical harm on their mates [9,10] and this species has been used routinely as a model system to study the economics and genetics of sexual conflict over lifespan and reproduction [34–36].

A recent study by Lymbery & Simmons [37] generally supported the importance of male relatedness in mediating

male harm to females. They found that females housed with familiar brothers produced more offspring, suggesting that relatedness and familiarity among males act together to reduce male-induced harm to females. The beetles in the Lymbery & Simmons [37] study were provided with baker's yeast, which is not a common condition for a species that inhabits human grain storages and is often kept in the laboratory as a capital breeder that is aphagous in the adult stage. Furthermore, while *C. maculatus* beetles can technically ingest yeast, yeast consumption *per se* does not necessarily have a positive effect on longevity, fecundity or offspring production [38]. Therefore, we investigated the effect of male relatedness and familiarity in a large outbred and well-described population of *C. maculatus* (SI USA) that was not provided with yeast in the adult stage, which is in line with the recent evolutionary history of this species and this population. We conducted a fully crossed experiment with respect to male relatedness and familiarity, quantifying the lifetime reproductive success and lifespan of virgin females exposed to four different trios of males: (i) brothers kept together versus unrelated males kept together; and (ii) brothers kept apart versus unrelated males kept apart. If within-group male relatedness is indeed mediating sexual conflict in this system, then brothers should cause less harm than unrelated males resulting in higher relative fitness of females.

## 2. Methods

### (a) Study system

Seed beetles are common pest of stored legumes indigenous to Asia and Africa. Females lay their eggs on the surface of dried beans. Once the larvae hatch, they burrow into the bean and eclose as reproductively mature adults approximately 23–27 days later. *Callosobruchus maculatus* are facultatively aphagous, obtaining all the nutrients they require for survival and reproduction during the larval stage [39]. Adult feeding increases fecundity and longevity [39]. Early studies used a combination of yeast and sugar solutions, so it was difficult to disentangle the effect of the separate components in fitness-related traits. Ursprung *et al.* [38] found that sugar solution and water do increase fecundity and longevity, but there was no effect of yeast on these key life-history traits. Lymbery & Simmons [37] provided their study beetles with ad libitum access to yeast, but there was no obvious benefit in terms of fecundity or longevity, although the direct comparison is not possible because their study did not include standard aphagous conditions.

The study population was derived from an outbred South Indian stock population (SI USA) of *C. maculatus* originally obtained from C. W. Fox at the University of Kentucky, USA, and then subsequently moved to Uppsala University and finally to the American University of Paris three months prior to the first block of the experiment. The original SI USA stock population was collected from infested mung beans (*Vigna radiata*) in Tirunelveli, India, in 1979 [40] and maintained in our laboratory for over 100 generations prior to the start of these experiments. Both prior to and during the experiment, beetles were cultured exclusively on mung beans and kept at aphagy (no food or water) in climate chambers at 29°C, 50% relative humidity and a 12 : 12 h light : dark cycle. One meaningful advantage of this system is that the laboratory conditions closely resemble natural conditions, because these beetles have associated with dried legumes for thousands of years and their life history is adapted to life in a storage environment [41,42].

## (b) Establishing the four treatment groups

The experiment was carried out in two blocks. During both blocks, base populations of beetles were kept in 1 l jars with 150 g of mung beans, and approximately 250 newly hatched beetles were transferred to new jars with fresh beans every 24 days on a continual basis. From this base population, we established four different treatment groups that differed in relatedness and social context (with a goal of approximately $n = 75$ each in each block, 150 total): (i) related kept in group (RG), (ii) non-related kept in group (NG), (iii) related kept alone (RA) and (iv) non-related kept alone (NA). In the RG treatment, three full-sib brothers were housed in a Petri dish together for 24 h before being added to a dish with a (non-related) female. In the NG treatment, three non-related males were housed together for 24 h before being added to a dish with a female. In the RA treatment, three brothers with no prior experience with each other were added all at once to a dish with a female. Finally, in the NA treatment, three non-related and unfamiliar males were added to a dish with a female.

To generate full sibling brothers for the RG and RA treatments, we transferred a random subset of beans with developing larvae into virgin chambers (aerated plastic culture plates with a separate well for each individual) and monitored the virgin chambers daily. Approximately 1 day after hatch, we randomly paired 180 males and females and placed them into 180 60 mm Petri dishes with 75 beans each. We then removed the males and females after 48 h and allowed the eggs to develop. Since females can lay up to 65 eggs per day (E.C.B. 2010, unpublished data), we wanted to provide enough beans that females would lay only one egg on each bean. Before the offspring hatched, we transferred the fertilized beans from the Petri dishes to virgin chambers, carefully marking which beans came from which parents, and monitoring hatch daily. Once these offspring hatched, we set up the four different treatment groups above.

For both 'group' treatments, trios of males were introduced to each other on the same day that they hatched, 24 h before being introduced to the female. For both 'alone' treatments, males were housed separately in individual virgin cells until 1 day post-hatch and then introduced together with the female. In all treatments, females were randomly selected from the base population 1 day after hatch and were unrelated to the males. Males from the non-related treatments were randomly selected from the base population as well. All sets of brothers used in the related treatments came from different parents, thus obviating the need to control for parental identity in the analyses. All males and females used in this study were maintained as virgins prior to the pairing. The Petri dishes in which males and females were housed measured 100 mm and contained 150 beans. This number is sufficient to allow females to lay just one egg per bean, reducing any larval competition that might affect data on reproductive success. This also means that there are no interactions between larvae in this study system.

The most natural way for *C. maculatus* males to interact with other beetles is when adults eclose and start mating straight away. Thus, the most natural set-up for a mating experiment is the one where individually raised beetles are grouped in either kin or non-kin groups, as in our 'alone' treatment. It is not impossible to imagine that, in some cases, male beetles could spend some of their time post-eclosion in the company of other male beetles before they encounter a female. While such a situation would be much rarer, we decided to model it as well, and introduced the 'group' treatment. In our experience with this study system that spans over a decade of research, it is unlikely that male beetles will ever spend longer than 24 h together before encountering a female, which is why we chose 24 h for the group treatment. It is important to note here that we are not specifically interested in comparing familiar kin with unfamiliar non-kin, because such a comparison is biologically not very meaningful here. The only biologically meaningful comparisons, in this system, are between kin and non-kin immediately after eclosion (normal scenario, 'alone' treatment), and between kin and non-kin after 24 h spent together in equal densities (rarer scenario, 'group' treatment).

Three days after the trios of males were introduced to females, we swapped out all the males for new males. This was done to reduce the variance caused by male condition or behaviour on female reproductive success or lifespan. The swapping was done just once because reproduction essentially ceases after 6 days. In preparation for this, for the group treatments, we set up new trios of freshly hatched related and non-related males 1 day before. For all the 'related' dishes, we used brothers of the previous trio. Since fewer males were eclosing this late in the hatch cycle, we had to use slightly older males in some cases. We excluded the few females that escaped/died from unnatural causes resulting in slight deviations from the initial sample size ($n = 75$ for RG, $n = 75$ for RA, $n = 71$ for NG and $n = 76$ for NA in the first block; $n = 75$ for all treatments in the second block).

## (c) Lifespan and fitness assays

During both blocks of the experiment, we conducted both lifespan and fitness assays for each female within each treatment group. For lifespan assays, we monitored the Petri dishes daily and recorded the date of death of each female. Once all adults were dead, we removed them from the dishes. We collected two kinds of fitness data. During the first block of the experiment, we measured total offspring production only. We did this by counting the number of eclosed young per female, a standard measure of lifetime reproductive success in this system. To facilitate the counting of offspring, we froze the dishes 37 days after the initial pairing, well after all the offspring had eclosed but before a subsequent generation could develop.

During the second block, we also measured daily offspring production for each female. To do this, we moved the female and males to new Petri dishes with new beans every 24 h until the female died (maximum of 9 sets of Petri dishes per female). Approximately 37 days later, we froze the dishes and counted the number of eclosed offspring per day per female.

## (d) Statistical analyses

Before analysis, we excluded all individuals that did not reproduce (NG = 2, NA = 4, RG = 4, RA = 10). We analysed the lifetime offspring production as well as age-specific reproduction using a generalized mixed effect model with a Poisson error structure implemented in the *lme4* package [43] in R v. 3.3.3., treating relatedness and social context as crossed fixed factors. We tested for overdispersion using the *dispersion_glmer* function in the *blmeco* package [44], and if above 1.4, we controlled for overdispersion by adding a subject-level random effect. For total reproduction, we used block as a random factor.

Age-specific reproduction and individual fitness was only analysed for block 2, as this was the only block where age-specific fecundity data were collected. For age-specific reproduction, we included relatedness and social context as crossed fixed factors, as well as all interactions with age and age$^2$. In addition, we also included age at last reproduction (ALR) as a crossed covariate. Age and ALR were scaled and centered before analysis (mean = 0, s.d. = 1) and we used the *bobyqa* optimizer (included in *lme4*) as well as increased the default number of iterations to 10 000 in order to obtain good model convergence. For all mixed-effect models, $\chi^2$ tests of fixed effects were performed using the *car* package [45].

Individual fitness ($\lambda_{ind}$) was calculated from the life-table of age-specific reproduction [46,47], with a development time of 23 days, by solving the Euler–Lotka equation for each individual

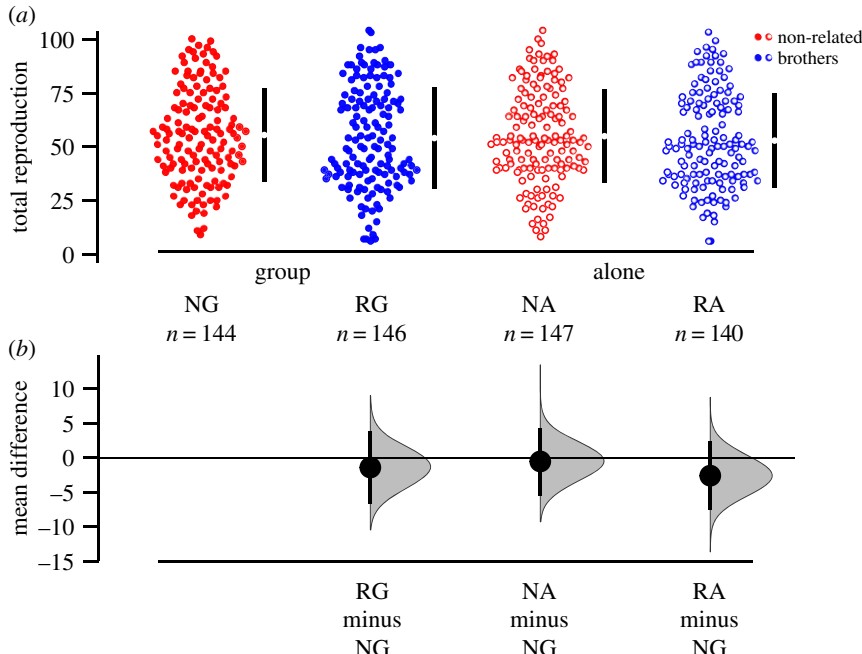

**Figure 1.** Lifetime reproductive success (LRS) (number of adult offspring) by treatment group: brothers (blue), non-related males (red), familiar individuals kept in group (solid symbols) and unfamiliar individuals kept alone (open symbols). (a) Raw data, with the mean ± 95% CI indicated with black bars at each group. (b) An estimation plot, where treatment levels are compared to NG, with a graded sampling distribution of bootstrapped values with a 95% CI.

using the *lambda* function in the *popbio* package [48]. $\lambda_{ind}$ was then analysed in a linear model using relatedness and social context as crossed fixed factors.

Survival was analysed in a Cox proportional hazard model using the *coxme* package [49], with relatedness and social context as crossed fixed factors, and block as a random effect.

ALR was investigated in a Cox proportional hazard model using the *coxph* function in the *survival* package [50], with relatedness and social context as crossed fixed factor. ALR was only scored in block 2.

Furthermore, we present our results using the most recent developments in data analysis and presentation [51], showing (i) the raw data as beehive plots; (ii) a summary of the data; and (iii) the result of bootstrapping analyses. These plots combine visual clarity with the statistical evaluation of the data.

## 3. Results

We measured lifetime reproductive success (total number of eclosed offspring) of individual females introduced to one of four different groups of male trios: brothers raised together (related group, or RG, $n = 146$), brothers raised separately (related alone, RA, $n = 140$), non-related males raised together (non-related group, NG, $n = 144$) or non-related males raised separately (non-related alone, NA, $n = 147$). There was no significant difference in female lifetime reproductive success among the four treatments (relatedness: $\chi^2 = 1.37$, d.f. = 1, $p = 0.392$; social context: $\chi^2 = 0.00$, d.f. = 1, $p = 0.999$; relatedness × social context: $\chi^2 = 0.0015$, d.f. = 1, $p = 0.969$; figure 1). If anything, the mean number of eclosed young was slightly higher for the non-related treatments. If the non-reproducing females are included in the dataset, we actually find higher reproduction in the non-related treatment group (relatedness: $\chi^2 = 4.30$, d.f. = 1, $p = 0.038$; social context: $\chi^2 = 1.33$, d.f. = 1, $p = 0.248$; relatedness × social context: $\chi^2 = 0.240$, d.f. = 1, $p = 0.624$; electronic supplementary material, figure S1). We did find different shapes of the age-specific fecundity, illustrated

by the significant interaction relatedness × social context × age$^2$ (table 1 and figure 2). However, we found no effect on individual fitness $\lambda_{ind}$ (relatedness: $F = 0.026$, d.f. = 1, $p = 0.872$; social context: $F = 0.244$, d.f. = 1, $p = 0.622$; relatedness × social context: $F = 0.072$, d.f. = 1, $p = 0.789$; figure 3).

When we measured the lifespan of the females introduced to the different treatment groups, we found that male relatedness improved female survival (table 2 and figure 4). By contrast, ALR was not influenced by relatedness or social context (table 2).

## 4. Discussion

Adaptive reduction in mate harm can only evolve if (i) there are reliable mechanisms for recognizing kin and (ii) populations are sufficiently viscous (i.e. genetically structured) for relatives to have a reasonable chance of encountering each other while they are reproductively active. The population genetic structure is one challenge facing the hypothesis that kin selection can mitigate the evolution of male harm via interlocus sexual conflict. For example, while kin recognition mechanisms may exist in *Drosophila* (e.g. through cuticular hydrocarbons [52,53] or gut microbiota [54,55]), Chippindale *et al.* [28] point out that both natural and laboratory populations of *Drosophila* are unlikely to be sufficiently structured to promote kin-selected reduction in male–female conflict. Simply put, adults emerge and fly off and are unlikely to remain in or disperse into genetically structured populations. Le Page *et al.* [32] countered this point by suggesting that genetic structure may occur in fruit flies during colonization of new patches by a small group of females, such that male relatedness-driven reduction in female harm in the established populations are a relic of 'the foundation past'. Future work will test whether selection during the foundation of a new population in the natural environment is sufficiently strong to generate long-lasting effects on male reproductive behaviour.

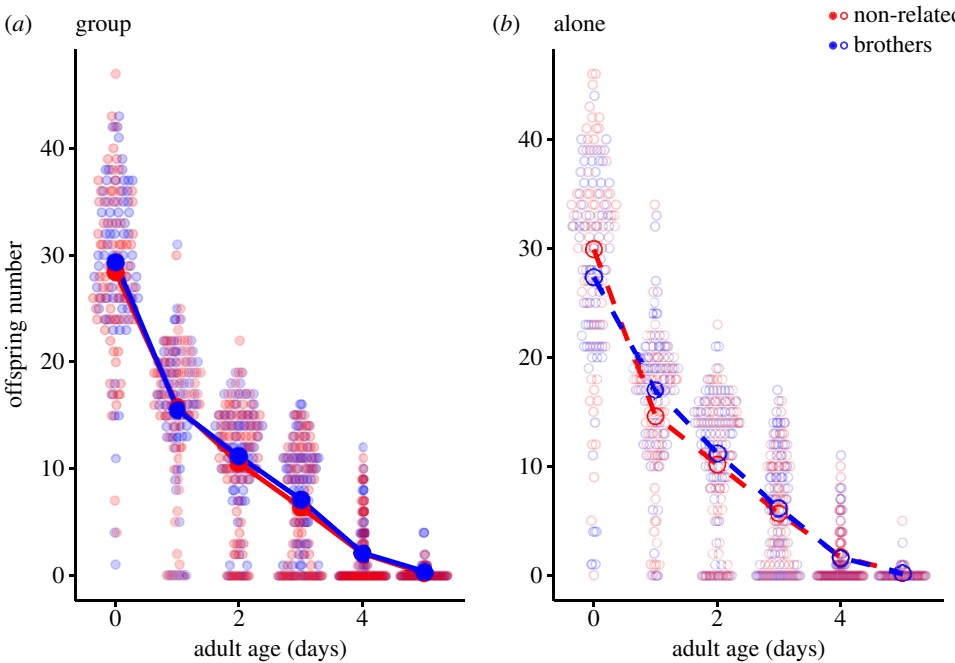

**Figure 2.** Age-specific reproduction for females mated with (*a*) males kept in group and (*b*) males kept alone. Brothers are shown as blue, and non-related males as red. Strong colour represents treatment means, and pale symbols show raw data.

**Table 1.** Age-specific reproduction, result from generalized linear model with poisson error structure.

| parameter | $\chi^2$ | d.f. | *p*-value |
|---|---|---|---|
| relatedness | 0.2719 | 1 | 0.60204 |
| social context | 0.1325 | 1 | 0.71588 |
| ALR | 52.8875 | 1 | <0.001 |
| age | 1459.7816 | 1 | <0.001 |
| age$^2$ | 623.6481 | 1 | <0.001 |
| relatedness × social context | 0.0653 | 1 | 0.79830 |
| relatedness × ALR | 0.2163 | 1 | 0.64190 |
| social context × ALR | 0.0061 | 1 | 0.93791 |
| relatedness × age | 1.3663 | 1 | 0.24246 |
| social context × age | 1.0830 | 1 | 0.29802 |
| ALR × age | 214.4389 | 1 | <0.001 |
| relatedness × age$^2$ | 1.6252 | 1 | 0.20237 |
| social context × age$^2$ | 1.8441 | 1 | 0.17448 |
| ALR × age$^2$ | 64.3620 | 1 | <0.001 |
| relatedness × social context × ALR | 1.6448 | 1 | 0.19967 |
| relatedness × social context × age | 0.9137 | 1 | 0.33914 |
| relatedness × ALR × age | 0.0013 | 1 | 0.97119 |
| social context × ALR × age | 2.4446 | 1 | 0.11793 |
| relatedness × social context × age$^2$ | 4.7754 | 1 | 0.02887 |
| relatedness × ALR × age$^2$ | 0.5634 | 1 | 0.45291 |
| social context × ALR × age$^2$ | 2.1438 | 1 | 0.14315 |
| relatedness × social context × ALR × age | 1.8233 | 1 | 0.17692 |
| relatedness × social context × ALR × age$^2$ | 3.3245 | 1 | 0.06825 |

By contrast, *C. maculatus* beetles may, in theory, meet the necessary pre-conditions without difficulty. Female seed beetles lay eggs in clusters, and will deposit all of their eggs in close proximity to each other, provided there is sufficient supply of unoccupied beans. Upon emergence from the bean, male seed beetles aggressively court females and begin mating immediately, which increases the probability of encountering relatives. *Callosobruchus maculatus* is a pest species that infests supplies

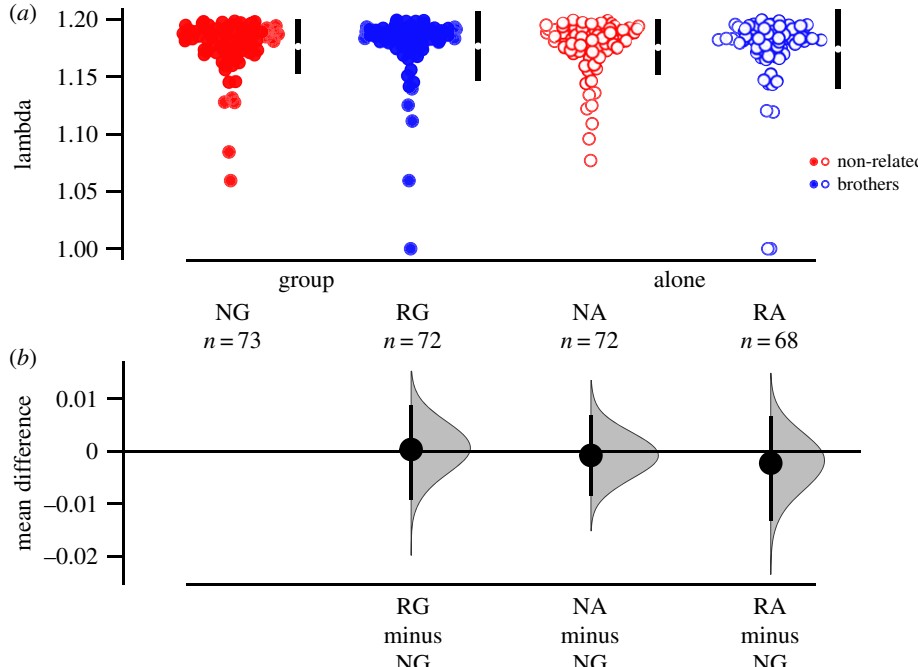

**Figure 3.** Individual fitness ($\lambda_{\mathrm{ind}}$) calculated from age-specific fecundity data, separated by treatment group: brothers (blue), non-related males (red), males kept in group (solid symbols) and males kept alone (open symbols). (a) Raw data, with the mean ± 95% CI indicated with black bars at each group. (b) An estimation plot, where treatment levels are compared to NG, with a graded sampling distribution of bootstrapped values with a 95% CI.

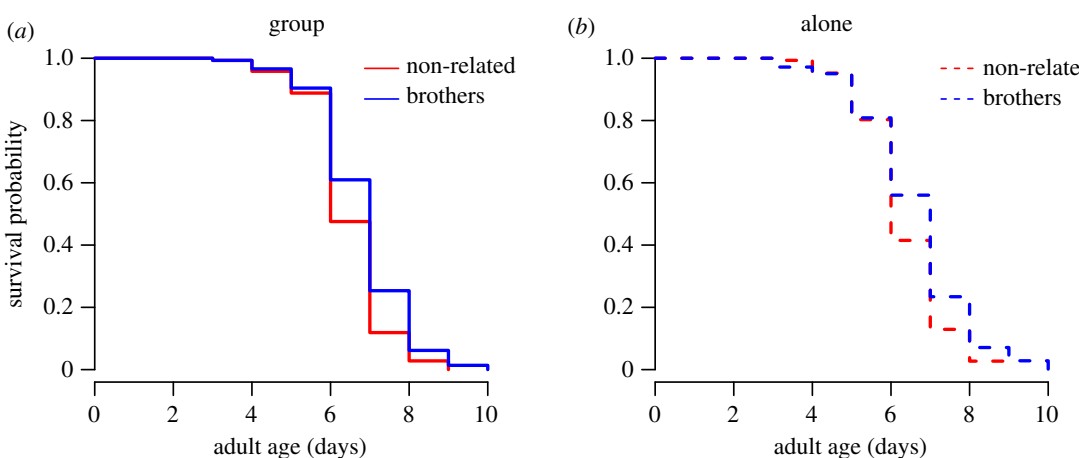

**Figure 4.** Survival probability for females mated to (a) males kept in group and (b) males kept alone. Blue represents brothers and red represents non-related males.

**Table 2.** Lifespan and ALR of females in response to male relatedness and social context.

| response | factor | coef | s.e. | z | *p*-value |
|---|---|---|---|---|---|
| lifespan | relatedness | −0.272 | 0.116 | −2.35 | 0.019 |
| | social context | 0.037 | 0.115 | 0.32 | 0.750 |
| | relatedness × social context | 0.036 | 0.163 | 0.22 | 0.820 |
| ALR | relatedness | −0.165 | 0.164 | −1.01 | 0.310 |
| | social context | 0.035 | 0.163 | 0.22 | 0.830 |
| | relatedness × social context | 0.215 | 0.231 | 0.93 | 0.350 |

of stored legumes—under those conditions, it is likely that many beetles will emerge and mate simultaneously providing sufficient variation in the relatedness of competitors. At the same time, we note that *C. maculatus* males are relatively indiscriminate in their mating behaviour, probably because of the high

'missing opportunity' cost that is associated with living in high-density populations, and commonly mount other males because of perception errors [56,57], which would complicate selection for a fine-tuned kin recognition mechanism that could lead, in theory, to reduced female harm.

In this study, similar to Chippindale et al.'s [28] Drosophila study, we found that neither male relatedness nor social context influenced female lifetime reproductive success. However, in contrast with Chippindale et al. [28], male relatedness improved female survival. This is also in contrast with Lymbery & Simmons [37], who found that both familiarity and relatedness increase reproductive success, but not survival. In our study, since the effect on survival occurred only in late life, around 6 days of age when females already stopped producing eggs, it failed to increase female lifetime reproductive success. Therefore, our results do not provide support for the role of kin selection in mitigating the effects of male harm.

Seed beetles are facultatively aphagous—that is, eclosed adults do not require food or water to breed and survive [58]. While in many bruchid beetles, adults commonly consume pollen, nectar or fungi [59], Callosobruchus beetles do not usually feed as adults. In the current study, we opted to keep the beetles under the aphagous conditions in which they have evolved for over 500 generations since they were first brought into the laboratory in 1979. Thus, our schedule reflects not only the original conditions of the human grain storage, but also the recent evolutionary history of this large outbred population.

Nevertheless, it is interesting to consider how male relatedness could affect the fitness of female beetles in different environments. One may hypothesize that increased survival could result in increased fecundity when beetles have access to additional resources to continue reproduction in late life. Yet, in a recent study by Lymbery & Simmons [37], beetles were raised with the access to yeast but neither lived longer nor produced more offspring compared to normal non-feeding conditions. This finding seems concordant with earlier reports that did not find positive effects of yeast consumption on fitness in C. maculatus [38]. However, despite the lack of positive effects of access to yeast on life-history traits, females kept with related familiar males produced more offspring in the Lymbery & Simmons [37] study. In our study, we found a three-way interaction between relatedness, social context and the shape of age-specific reproduction curve, stemming from the increased early-life reproduction and steeper age-specific decline of females mated to unrelated males than females mated to groups of brothers raised alone. Increased reproductive performance in early-life could translate into increased individual fitness, but this was not the case. On the other hand, when we included females that failed to produce viable offspring in the analysis, we found that females kept with groups of unrelated males had higher reproductive success, because most of failed reproductive attempts were among females kept with groups of brothers. This finding is in line with the idea that multiple mating increases female fitness when it increases the genetic diversity of partners [60–63]. Indeed, D. melanogaster females re-mate more often when facing a group of genetically diverse males [64].

To summarize, there is little conclusive evidence to date for the role of kin selection in mediating sexual conflict, and, specifically, in reducing male-induced harm to females. More importantly, it is not always easy to see how the selection for such an effect can operate in the natural environment because it requires many opportunities for sib–sib interaction that may not be very common in wild populations of invertebrates [28]. Le Page et al. [32] discussed several possible explanations for the fact that reduction in male-induced harm to females is observed in some populations. One likely non-adaptive explanation is a perception error, first used in this context by Hollis et al. [29]. Indeed, males can use cuticular hydrocarbon profiles or gut microbiota as a measure of male–male competition, and they may underestimate the level of competition when surrounded exclusively by related males with similar odours, thereby investing less in sperm competition. Such a non-adaptive hypothesis fits squarely with the results of our study, because females housed with groups of related males did enjoy improved survival, suggesting reduced male harm, but this effect was entirely limited to the post-reproductive part of their life cycle and had no effect on their individual fitness. We suggest that more work is needed to evaluate the importance of within-group male relatedness in the evolution of mating systems.

Data accessibility. The complete dataset and all R scripts are provided in the electronic supplementary material.

Authors' contributions. E.C.B. designed the experiment, helped collect the data and co-wrote the manuscript. M.I.L. conducted the statistical analyses and co-wrote the manuscript. S.M. helped collect the data and co-wrote the manuscript. S.B. helped collect the data. A.A.M. conceived of the initial idea, helped design the experiment and co-wrote the manuscript.

Competing interests. We declare we have no competing interests

Funding. ERC Starting Grant 2010 AGINGSEXDIFF and ERC Consolidator Grant 2017 GERMLINEAGEINGSOMA to A.A.M., Swedish Research Council VR grant no. 2016-05195 to M.I.L. and funding from the American University of Paris to E.C.B. supported this study.

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
