## [Reviewer comments · Proceedings of the Royal Society B: Biological Sciences]

Review History

RSPB-2019-1137.R0 (Original submission)

Review form: Reviewer 1

Recommendation

Major revision is needed (please make suggestions in comments)

Scientific importance: Is the manuscript an original and important contribution to its field?

Good

General interest: Is the paper of sufficient general interest?

Good

Quality of the paper: Is the overall quality of the paper suitable?

Good

Is the length of the paper justified?

Yes

Should the paper be seen by a specialist statistical reviewer?

No

Do you have any concerns about statistical analyses in this paper? If so, please specify them explicitly in your report.

No

It is a condition of publication that authors make their supporting data, code and materials available - either as supplementary material or hosted in an external repository. Please rate, if applicable, the supporting data on the following criteria.

Is it accessible?

Yes

Is it clear?

Yes

Is it adequate?

Yes

Do you have any ethical concerns with this paper?

No

Comments to the Author

In this paper Berg et al. tests if kin selection mediates sexual conflict in the seed beetles as proposed by Carazo et al. (2014) using *Drosophila melanogaster*. Overall this is a well-written and valuable paper and it is nice to see this idea tested in a species other than fruit flies. The authors also do a well job in explaining why this is good species and setup for exploring this phenomenon.

I have only one major concern about this study. As far as I can understand (but I might be mistaken) in the familiarity treatments three males (related or unrelated) were kept together for 24 hours after eclosion, and in the unfamiliar treatments males were held individually. I have two concerns about this setup:

1) The fact that the unfamiliar ones were held individually for 24 hours rather than in a group of three makes the design unbalanced. This might have introduced variation that is not explained by familiarity but by the beetles' perception of sperm competition. The design would have been much better if the unfamiliar ones were also kept in a group of three for 24 hours but then grouped with other individuals when exposed to the female.

2) I am also not convinced that holding three beetles together for 24 hours after eclosion creates familiarity. The fruit fly experiments that this work is based upon create familiarity at the larval stage so that the flies are 'raised' together until they eclose. This could perhaps have been achieved here if beans from different families (in larval stage) were put together in a petri dish and collected after eclosion. In the current setup I am not sure if 24 hours together after eclosion can be categorised as 'raised' together and would indicate familiarity.

I generally think that these shortcomings have to be acknowledged and discussed in the manuscript, especially because they might influence the outcome of the experiment. This is a minor point but Figure 1 (and maybe Figure 2) would look better if they include the data points as well.

Review form: Reviewer 2

Recommendation

Major revision is needed (please make suggestions in comments)

Scientific importance: Is the manuscript an original and important contribution to its field?

Good

General interest: Is the paper of sufficient general interest?

Good

Quality of the paper: Is the overall quality of the paper suitable?

Acceptable

Is the length of the paper justified?

Yes

Should the paper be seen by a specialist statistical reviewer?

No

Do you have any concerns about statistical analyses in this paper? If so, please specify them explicitly in your report.

No

It is a condition of publication that authors make their supporting data, code and materials available - either as supplementary material or hosted in an external repository. Please rate, if applicable, the supporting data on the following criteria.

Is it accessible?

No

Is it clear?

No

Is it adequate?

No

Do you have any ethical concerns with this paper?

No

Comments to the Author

The manuscript RSPB-2019-1137 by Berg et al. examines whether the degree of male-male interaction in the seed beetle *C. maculatus* is mediated by relatedness between individual males and/or their familiarity. The role of kinship in sexual selection / sexual conflict is a highly debated topic, and in need of more empirical studies that can weigh in on its importance and universality. Overall, I thought the experiment was well designed & executed, enjoyed reading this manuscript, but I did have some questions and concerns that I hope the authors will be able to address. I have tried to

Broader questions:

After having read the passage in the introduction (lines 124-135), the methods (lines 160-1623) and the discussion (lines 342 -344, 362-370) I am still not 100% clear whether the authors are arguing that Lymbery et al.'s study is confounded by the presence of Baker's yeast, especially as

they don't require eating as adults (lined 349). Surely there are other differences that could also contribute? How do you reconcile these results? Can it be reconciled?

Line 275: Since in block 2 you measured daily female egg production, could you could also have performed a survival analysis on the ALR as well? This would be relevant to your discussion.

Line 396- end: I felt that your perception error could be better developed. It was quite brief.

Minor points:

Line 67: Here, "viscous" should be defined as having genetic structure, otherwise the interpretation is more open.

Line 160-163: This phrasing is odd. Your statement says there is no obvious benefit to yeast, but it is also impossible to tell if there was no benefit to yeast.

Lines 166. When did the population move to Uppsala?

Line 173: 'meaningful' instead of 'great'

Line 222. I think it is worth pointing out that males were only swapped out once @ day 3 rather than multiply (as was done in some of the fly work), as the experiment lasted at most 10 days

Line 257: Please add in citations for the non-base R packages you used (such as blmeco, popbio, car)

Line 259: Please consider including all your R scripts in the supplementary files along with your archived data.

Line 266: Is your bobyqa optimizer a function in a package, or is there some other source of this? Please cite.

Line 288-290. To back up this 'if anything' statement, perhaps an analysis that only uses relatedness? What about an effect size statistic to better describe this difference (if there is one)?

Line 299-303: While you provide the Wald statistic (z) for the beta coefficients, it would be more beneficial to provide the hazard ratios (& 95% CIs) to give the effect sizes associated with relatedness and/or familiarity?

Line 313: Kin recognition in fruit flies may also depend on microbiome cues:

Lizé, A., McKay, R., & Lewis, Z. (2014). Kin recognition in *Drosophila*: the importance of ecology and gut microbiota. *The ISME journal*, 8(2), 469.

Lizé, A., McKay, R., & Lewis, Z. (2013). Gut microbiota and kin recognition. *Trends in ecology & evolution*, 28(6), 325-326.

Lewis, Z., Heys, C., Prescott, M., & Lizé, A. (2014). You are what you eat: gut microbiota determines kin recognition in *Drosophila*. *Gut microbes*, 5(4), 541-543.

Line 382: Diversity in mates increases the chance of genetic compatibility

Figure1 (& S1): Mean +/- SE plots are not as meaningful as boxplots or violin plots as describing your data.

Finally: Your format of your Raw data as a pdf of an excel sheet is not accessible, and .csv files

should be uploaded as ESM.

Decision letter (RSPB-2019-1137.R0)

11-Jun-2019

Dear Dr Berg:

We are writing to inform you that we have now obtained responses from referees on manuscript RSPB-2019-1137 entitled "Kin selection and sexual conflict: male relatedness and familiarity do not affect female fitness in seed beetles" which you submitted to Proceedings B.

Unfortunately, on the advice of the Associate Editor and the referees, your manuscript has been rejected following full peer review. Competition for space in Proceedings B is currently extremely severe, as many more manuscripts are submitted to us than we have space to print. We are therefore only able to publish those that are exceptional, convincing and present significant advances of broad interest, and must reject many good manuscripts.

On a more positive note, based on the advice we have received, we would like to offer you the opportunity to transfer your manuscript file to another Royal Society journal, Royal Society Open Science. Royal Society Open Science is a fast, open journal publishing high-quality research across all of science and mathematics. The journal operates objective peer review, optional open peer review, and will publish any article deemed to sufficiently advance the field by the reviewers and editors, leaving judgement of potential impact of the work to the reader. The journal publishes Registered Reports and encourages the submission of negative results. You can find out more about the scope of the journal and the benefits of publication here <https://royalsocietypublishing.org/journal/rsos>

If you wish to have your manuscript transferred to Royal Society Open Science please ensure that you revise your text to address all of the reviewers' comments relating to scientific soundness. Please particularly ensure that your conclusions do not overstate the results of your study. Once submitted to Royal Society Open Science your manuscript will be assessed by an Associate Editor who will decide whether further reviewer advice is required. If no further advice is needed and all of your revisions are satisfactory your manuscript will be immediately accepted for publication.

If you agree to transfer your paper, and it is accepted for publication, you will be asked to pay the article processing charge, unless you request a waiver and this is approved by Royal Society Publishing. You can find out more about the charges at <https://royalsocietypublishing.org/rsos/charges>.

You can approve or reject this transfer using the links below:

Approve transfer - *** PLEASE NOTE: This is a two-step process. After clicking on the link, you will be directed to a webpage to confirm. ***

https://mc.manuscriptcentral.com/prsb?URL_MASK=64dc5e85ef3842f3a15a9de22195dc0b

After approving the transfer you will need to log in to your Royal Society Open Science author centre (<https://mc.manuscriptcentral.com/rsos>) to complete your the submission. At this stage you will have chance to address any of the reviewers' or editor's concerns.

Reject transfer - *** PLEASE NOTE: This is a two-step process. After clicking on the link, you will be directed to a webpage to confirm. ***

https://mc.manuscriptcentral.com/prsb?URL_MASK=6cc64ecc75bc4584bc2d359027b3d697

or by clicking 'approve' or 'reject' in your Author Center.

Once you have approved the transfer you will be prompted to complete the transfer of your article via the Royal Society Open Science submission system.

Please find below the comments received from referees concerning your manuscript, not including confidential reports to the Editor. If you approve transfer to Royal Society Open Science, these reviews will accompany your paper.

Thank you for your interest in Proceedings B.

Sincerely,
Proceedings B
mailto:proceedingsb@royalsociety.org

=====

Associate Editor, Comments to Author:

This manuscript attempts to test the hypothesis that kin selection can reduce sexual conflict adaptations using a factorial design in seed beetles. They found a marginal increase in female survival when housed with related males, but since this increase mainly influences the post-reproductive lifespan, it did not result in increased lifetime reproductive success. They therefore conclude that kin selection is not an important force in this system.

This manuscript is one of a number of recent studies investigating the influence of kin selection on sexual conflict, and as such its novelty is perhaps not the highest. There are also some issues with the experimental design that cast some doubt on the robustness of the results. For one thing, the "familiar" males were only kept together for 24 hours, so it seems that an even better test of familiarity vs. genetic relatedness would use males that had grown up together in the same jar. In addition, as pointed out by reviewer 1, since the "unfamiliar" males were kept individually, differences in how the males affected females could be a result of altered investment due to differences in apparent sperm competition risk rather than familiarity per se. Although otherwise technically sound, I think these issues reduce the impact of the paper, making it more appropriate for publication in Royal Society Open Science.

Reviewers' Comments to Author:

Referee: 1

In this paper Berg et al. tests if kin selection mediates sexual conflict in the seed beetles as proposed by Carazo et al. (2014) using *Drosophila melanogaster*. Overall this is a well-written and valuable paper and it is nice to see this idea tested in a species other than fruit flies. The authors also do well in explaining why this is a good species and setup for exploring this phenomenon.

I have only one major concern about this study. As far as I can understand (but I might be mistaken) in the familiarity treatments three males (related or unrelated) were kept together for 24 hours after eclosion, and in the unfamiliar treatments males were held individually. I have two concerns about this setup:

1) The fact that the unfamiliar ones were held individually for 24 hours rather than in a group of three makes the design unbalanced. This might have introduced variation that is not explained by familiarity but by the beetles' perception of sperm competition. The design would have been much better if the unfamiliar ones were also kept in a group of three for 24 hours but then grouped with other individuals when exposed to the female.

2) I am also not convinced that holding three beetles together for 24 hours after eclosion creates familiarity. The fruit fly experiments that this work is based upon create familiarity at the larval stage so that the flies are 'raised' together until they eclose. This could perhaps have been achieved here if beans from different families (in larval stage) were put together in a petri dish and collected after eclosion. In the current setup I am not sure if 24 hours together after eclosion can be categorised as 'raised' together and would indicate familiarity.

I generally think that these shortcomings have to be acknowledged and discussed in the manuscript, especially because they might influence the outcome of the experiment. This is a minor point but Figure 1 (and maybe Figure 2) would look better if they include the data points as well.

Referee: 2

The manuscript RSPB-2019-1137 by Berg et al. examines whether the degree of male-male interaction in the seed beetle *C. maculatus* is mediated by relatedness between individual males and/or their familiarity. The role of kinship in sexual selection / sexual conflict is a highly debated topic, and in need of more empirical studies that can weigh in on its importance and universality. Overall, I thought the experiment was well designed & executed, enjoyed reading this manuscript, but I did have some questions and concerns.

Broader questions:

After having read the passage in the introduction (lines 124-135), the methods (lines 160-1623) and the discussion (lines 342 -344, 362-370) I am still not 100% clear whether the authors are arguing that Lymbery et al.'s study is confounded by the presence of Baker's yeast, especially as they don't require eating as adults (lined 349). Surely there are other differences that could also contribute? How do you reconcile these results? Can it be reconciled?

Line 275: Since in block 2 you measured daily female egg production, could you could also have performed a survival analysis on the ALR as well? This would be relevant to your discussion.

Line 396- end: I felt that your perception error could be better developed. It was quite brief.

Minor points:

Line 67: Here, "viscous" should be defined as having genetic structure, otherwise the interpretation is more open.

Line 160-163: This phrasing is odd. Your statement says there is no obvious benefit to yeast, but it is also impossible to tell if there was no benefit to yeast.

Lines 166. When did the population move to Uppsala?

Line 173: 'meaningful' instead of 'great'

Line 222. I think it is worth pointing out that males were only swapped out once @ day 3 rather than multiply (as was done in some of the fly work), as the experiment lasted at most 10 days

Line 257: Please add in citations for the non-base R packages you used (such as blmeco, popbio, car)

Line 259: Please consider including all your R scripts in the supplementary files along with your archived data.

Line 266: Is your bobyqa optimizer a function in a package, or is there some other source of this? Please cite.

Line 288-290. To back up this 'if anything' statement, perhaps an analysis that only uses relatedness? What about an effect size statistic to better describe this difference (if there is one)?

Line 299-303: While you provide the Wald statistic (z) for the beta coefficients, it would be more beneficial to provide the hazard ratios (& 95% CIs) to give the effect sizes associated with relatedness and/or familiarity?

Line 313: Kin recognition in fruit flies may also depend on microbiome cues:

Lizé, A., McKay, R., & Lewis, Z. (2014). Kin recognition in *Drosophila*: the importance of ecology and gut microbiota. *The ISME journal*, 8(2), 469.

Lizé, A., McKay, R., & Lewis, Z. (2013). Gut microbiota and kin recognition. *Trends in ecology & evolution*, 28(6), 325-326.

Lewis, Z., Heys, C., Prescott, M., & Lizé, A. (2014). You are what you eat: gut microbiota determines kin recognition in *Drosophila*. *Gut microbes*, 5(4), 541-543.

Line 382: Diversity in mates increases the chance of genetic compatibility

Figure1 (& S1): Mean +/- SE plots are not as meaningful as boxplots or violin plots as describing your data.

Finally: Your format of your Raw data as a pdf of an excel sheet is not accessible, and .csv files should be uploaded as ESM.

Author's Response to Decision Letter for (RSPB-2019-1137.R0)

See Appendix A.

RSPB-2019-1664.R0

Review form: Reviewer 2

Recommendation

Accept as is

Scientific importance: Is the manuscript an original and important contribution to its field?

Good

General interest: Is the paper of sufficient general interest?

Good

Quality of the paper: Is the overall quality of the paper suitable?

Good

Is the length of the paper justified?

Yes

Should the paper be seen by a specialist statistical reviewer?

No

Do you have any concerns about statistical analyses in this paper? If so, please specify them explicitly in your report.

No

It is a condition of publication that authors make their supporting data, code and materials available - either as supplementary material or hosted in an external repository. Please rate, if applicable, the supporting data on the following criteria.

Is it accessible?

Yes

Is it clear?

Yes

Is it adequate?

Yes

Do you have any ethical concerns with this paper?

No

Comments to the Author

I am very pleased to see that the authors have addressed clearly & completely all of the issues I raised in my first review of the manuscript.

Review form: Reviewer 3

Recommendation

Accept with minor revision (please list in comments)

Scientific importance: Is the manuscript an original and important contribution to its field?

Excellent

General interest: Is the paper of sufficient general interest?

Excellent

Quality of the paper: Is the overall quality of the paper suitable?

Excellent

Is the length of the paper justified?

Yes

Should the paper be seen by a specialist statistical reviewer?

No

Do you have any concerns about statistical analyses in this paper? If so, please specify them explicitly in your report.

No

It is a condition of publication that authors make their supporting data, code and materials available - either as supplementary material or hosted in an external repository. Please rate, if applicable, the supporting data on the following criteria.

Is it accessible?

Yes

Is it clear?

Yes

Is it adequate?

Yes

Do you have any ethical concerns with this paper?

No

Comments to the Author

This paper has undergone round(s) of reviews before I've seen it. Overall I think it is very good shape: solid work, well written and clear manuscript, with a largely balanced treatment of the field. I've only a few of comments to do with interpretation where I think pretty minor edits could potentially improve the ms. However, I recognise that some of these are relatively subjective: it is up to the authors and editor as to whether these are incorporated.

1) The life in storage environments, egg laying in clusters, and immediate mating upon eclosion are used to argue that *C. maculatus* are a ripe system for testing the kin selection mediated sexual conflict ideas. But presumably what really matters is viscosity, dispersal, and whether males can encounter socially variable environments in their lifetime (i.e. sometimes brothers, other times not)? I suspect the answer is that we don't know much about these things, just as we don't for

Drosophila either (or indeed most small insects). A note to that effect I think would balance things up.

2) I don't think the authors provided a particularly satisfactory response to Referee 2's first point, about the yeast difference in the Lymbery paper. The text in the main ms seems to be insinuating that there was something weird about the yeast treatment in the Lymbery study which somehow undermines it. It would be nice if this was toned down somewhat. I don't know enough about *C. maculatus* to say either way, but I can't see any obvious reason why one should believe the results of one study over the other. The statement on Line 399-401 "... the presence of live yeast interacts with male behaviour..." seems unjustified. They are different experiments on different beetles in different labs, so there could be loads of reasons for the differences. It is, superficially at least, hard to see how yeast could explain it.

3) Line 421-423 argue why bulb mites are particularly suited to the evolution of kin selected benefits, due to rapid population growth and colonisation of new patches. Isn't this true for lots of insects too? I don't obviously see how this argument makes bulb mites especially more suited than beetles or flies.

4) A recognition failure of kin does not automatically mean something non-adaptive. Recognition or not is about mechanisms not cost and benefits. For example, if females of a hypothetical species failed to recognise related brothers as potential mates, and therefore rejected them, we'd probably interpret that as a mechanism for adaptive inbreeding avoidance. I.e. there could be adaptive recognition failure. I do think its perfectly reasonable to make recognition failure a parsimonious mechanism, and also to suggest that its non-adaptive. But it's conceptually useful to articulate that one does not necessitate the other.

Decision letter (RSPB-2019-1664.R0)

29-Jul-2019

Dear Dr Berg:

Your manuscript has now been peer reviewed and the reviews have been assessed by an Associate Editor. The reviewers' comments (not including confidential comments to the Editor) and the comments from the Associate Editor are included at the end of this email for your reference. As you will see, the reviewers and the Editors have raised some concerns with your manuscript and we would like to invite you to revise your manuscript to address them.

When submitting your revision please upload a file under "Response to Referees" in the "File

Upload" section. This should document, point by point, how you have responded to the reviewers' and Editors' comments, and the adjustments you have made to the manuscript. We require a copy of the manuscript with revisions made since the previous version marked as 'tracked changes' to be included in the 'response to referees' document.

Research ethics:

Use of animals and field studies:

Please submit a copy of your revised paper within three weeks. If we do not hear from you within this time your manuscript will be rejected. If you are unable to meet this deadline please let us know as soon as possible, as we may be able to grant a short extension.

Best wishes,
Victoria Braithwaite

=====
Professor V A Braithwaite
mailto:proceedingsb@royalsociety.org
=====

Associate Editor Board Member, Comments to Author:

This ms is a resubmission the original submission has been handled by a different associate editor. It reports an experiment testing the idea that relatedness and/or familiarity among competing males reduces their harm to females. There is some, although not universal, support for this idea from studies in *Drosophila*, and from a recent paper in the seed beetle *Callosobruchus maculatus*. This paper reports an experiment on the latter species which find no such effect. Thus, while the present paper is not conceptually new, it is important in that its results go, in a way, against the grain. All too often publication bias against negative results creates an exaggerated impression of prevalence of a particular phenomenon.

The ms has been reviewed by two reviewers. Reviewer 1, who is one of the original reviewers, is completely satisfied with the revisions. Reviewer 2 is likewise positive, but brings up a few minor points related to some rather speculative arguments made by the authors, notably the attribution of the differences with a previous paper to yeast supplement or the speculations as to whether kin selection might operate in the seed beetle or in the bulb mite. The present paper provides no data to bear on these issues; the mention of the bulb mite could be cut completely, and while the yeast supplement difference should be mentioned, there is no basis for attributing the difference between studies to this protocol difference.

To those, I would like to add a few issues on my own:

(1) In the "familiar" treatment the three competing males were kept together for 24 h before being presented with the female whereas in the "unfamiliar" treatment the males were kept singly until being put in trios and immediately introduced to the female (l. 210-212). Thus it is not only that they are unfamiliar with their competitors; they have no experience of interaction with other males at all. At least in *Drosophila*, prior experience with rivals affects male courtship and mating behavior (several papers by Brettman). "Unfamiliar" is thus a misnomer for the treatment. If the authors really wanted to study the effect of familiarity, the "unfamiliar" males should have been also kept in trios for 24 h, but the trios should have been reshuffled before introducing the female.

This seems to have been done by Lymbery and Simmons, so this is another difference between the studies that should be mentioned. Thus, the text should be revised throughout to avoid conveying the false impression that it is the effect of familiarity that is being tested.

(2) While it is true that kin selection can be predicted to favor reduced sexual conflict when competing males are related, there are other, more plausible reasons for reduced male-male aggression or female harm when then competitors are brothers or why they are familiar with one another. The original Carazo et al paper has been criticized by several authors (ref 29-31) for making unsubstantiated claims about kin selection. The present ms does acknowledge some of these more parsimonious explanations in the discussion; however, the title, abstract and the introduction imply the experiment reported in this paper is somehow testing kin selection. It is not; whatever the results might have been, the experiments performed would not allow the authors to conclude anything about kin selection. This point is also raised by reviewer 2, who points out that the absence of kin recognition does not imply absence of kin selection; in the same way, differential behavior in the presence of kin versus non-kin would not imply kin selection. Therefore, there is no place for kin selection in the title. The abstract and introduction should be restructured to start with the phenomenon of reduced conflict if males are related and/or familiar, and the potential reasons should be briefly reviewed, including kin selection but without giving it undue weight.

(3) Related to the above, among the different alternative explanations, the authors focus in the discussion on the "perception error". This is certainly plausible, but a couple of other explanations deserve some discussion, as they are supported by data from *Drosophila*, notably that females re-mate more when facing a group of genetically more diverse males (Krupp et al 2008 *Current Biol.*) or that the degree of male harm to females is largely determined by the most aggressive male of the competing group (ref 31). Do we know anything about this in *C. maculatus*?

(4) The ms seems rather long, and the additions suggested above would make it even longer. However, there is a potential for tightening the text. Notably, the introduction gives a detailed chronological story of the different studies who found or did not find the focal effect in *Drosophila*, and this can be tightened. Similarly, it is quite intuitive why kin selection might reduce male-male aggression and female harm, do that part could also shortened.

Minor points

l. 99. Hollis et al 2015 explicitly set out to test if "familiarity" (or more precisely, rearing brothers together) was necessary for the reduced female harm in the context of the earlier Carazo et al paper. They did not ask if it was sufficient, and thus an unrelated-familiar treatment was not necessary. So referring to the absence of such a treatment as a weakness is being unnecessarily judgmental.

l. 317: "between the four treatments" - should be "among"?

l. 428: The "perception error" due to converged CHC as a potential explanation for reduced female harm when males are kept together was first proposed by Hollis et al 2015, and it would be fair to acknowledge this here.

=====

Reviewers' Comments to Author:

====

Referee: 2

I am very pleased to see that the authors have addressed clearly & completely all of the issues I raised in my first review of the manuscript.

===

Referee: 3

This paper has undergone round(s) of reviews before I've seen it. Overall I think it is very good shape: solid work, well written and clear manuscript, with a largely balanced treatment of the field. I've only a few of comments to do with interpretation where I think pretty minor edits could potentially improve the ms. However, I recognise that some of these are relatively subjective: it is up to the authors and editor as to whether these are incorporated.

1) The life in storage environments, egg laying in clusters, and immediate mating upon eclosion are used to argue that *C maculatus* are a ripe system for testing the kin selection mediated sexual conflict ideas. But presumably what really matters is viscosity, dispersal, and whether males can encounter socially variable environments in their lifetime (i.e. sometimes brothers, other times not)? I suspect the answer is that we don't know much about these things, just as we don't for *Drosophila* either (or indeed most small insects). A note to that effect I think would balance things up.

2) I don't think the authors provided a particularly satisfactory response to Referee 2's first point, about the yeast difference in the Lymbery paper. The text in the main ms seems to be insinuating that there was something weird about the yeast treatment in the Lymbery study which somehow undermines it. It would be nice if this was toned down somewhat. I don't know enough about *C maculatus* to say either way, but I can't see any obvious reason why one should believe the results of one study over the other. The statement on Line 399-401 "... the presence of live yeast interacts with male behaviour..." seems unjustified. They are different experiments on different beetles in different labs, so there could be loads of reasons for the differences. It is, superficially at least, hard to see how yeast could explain it.

3) Line 421-423 argue why bulb mites are particularly suited to the evolution of kin selected benefits, due to rapid population growth and colonisation of new patches. Isn't this true for lots of insects too? I don't obviously see how this argument makes bulb mites especially more suited than beetles or flies.

4) A recognition failure of kin does not automatically mean something non-adaptive. Recognition or not is about mechanisms not cost and benefits. For example, if females of a hypothetical species failed to recognise related brothers as potential mates, and therefore rejected them, we'd probably interpret that as a mechanism for adaptive inbreeding avoidance. I.e. there could be adaptive recognition failure. I do think its perfectly reasonable to make recognition failure a parsimonious mechanism, and also to suggest that its non-adaptive. But it's conceptually useful to articulate that one does not necessitate the other.

Author's Response to Decision Letter for (RSPB-2019-1664.R0)

See Appendix B.

Decision letter (RSPB-2019-1664.R1)

19-Aug-2019

Dear Dr Berg

I am pleased to inform you that your manuscript entitled "Kin but less than kind: within-group male relatedness does not increase female fitness in seed beetles" has been accepted for publication in Proceedings B.

Open Access

Paper charges

Sincerely,
Victoria Braithwaite

Professor V A Braithwaite
Editor, Proceedings B
<mailto:proceedingsb@royalsociety.org>

Appendix A

Response to reviewers:

Referee: 1

COMMENT: In this paper Berg et al. tests if kin selection mediates sexual conflict in the seed beetles as proposed by Carazo et al. (2014) using *Drosophila melanogaster*. Overall this is a well-written and valuable paper and it is nice to see this idea tested in a species other than fruit flies. The authors also do well in explaining why this is a good species and setup for exploring this phenomenon.

Response: Thank you very much for this.

COMMENT: I have only one major concern about this study. As far as I can understand (but I might be mistaken) in the familiarity treatments three males (related or unrelated) were kept together for 24 hours after eclosion, and in the unfamiliar treatments males were held individually. I have two concerns about this setup:

1) The fact that the unfamiliar ones were held individually for 24 hours rather than in a group of three makes the design unbalanced. This might have introduced variation that is not explained by familiarity but by the beetles'™ perception of sperm competition. The design would have been much better if the unfamiliar ones were also kept in a group of three for 24 hours but then grouped with other individuals when exposed to the female.

2) I am also not convinced that holding three beetles together for 24 hours after eclosion creates familiarity. The fruit fly experiments that this work is based upon create familiarity at the larval stage so that the flies are 'raised' together until they eclose. This could perhaps have been achieved here if beans from different families (in larval stage) were put together in a petri dish and collected after eclosion. In the current setup I am not sure if 24 hours together after eclosion can be categorised as 'raised' together and would indicate familiarity. I generally think that these shortcomings have to be acknowledged and discussed in the manuscript, especially because they might influence the outcome of the experiment.

Response: There is a critical difference between the biology of the seed beetle and the biology of the fruit fly, *Drosophila melanogaster*, that was used as a model system in the majority of the previous studies. *D. melanogaster* larvae spend a lot of time foraging together in the fly medium; during this time they interact, and these interactions can influence their subsequent behaviour and life-history decisions. *Callosobruchus maculatus* seed beetles used in our study originated from a population developing on mung beans in India in 1979, and have been raised on mung beans ever since. Female *C. maculatus* in this population lay one egg per bean; the hatched larvae burrow inside the bean and remain there for the rest of their development, while foraging alone, until eclosing as reproductively active adults. This life-history precludes interaction between larvae prior to eclosion. For this reason,

it is not possible to study the interactions between larvae in our study system. We thank Reviewer 1 for bringing this up and we hope this answers the main concern. We did mention this in the Methods section of the original manuscript but it was not sufficiently clear and we now make this more explicit in the text (Lines 232-233 of the “track changes” document).

The most natural way for *C. maculatus* to interact with other beetles is when adults hatch and start mating straight away. Thus, the most natural set-up of the experiment is one where individually raised beetles are grouped in either kin or non-kin groups, as in our Unfamiliar treatment. It is not impossible to imagine that, in some cases, male beetles could spend some of their time post-eclosion in the company of other male beetles before they encounter a female. While such a situation would be much rarer, we decided to model it as well, and introduced the Familiar treatment. In our experience with this study system that spans over a decade of research, it is unlikely that male beetles will ever spend longer than 24h together before encountering a female, which is why we chose 24h for the Familiar treatment. It is important to note here that we are not specifically interested in comparing, say Familiar Kin with Unfamiliar Non-kin, because such a comparison is biologically not very meaningful. The only biologically meaningful comparisons, in this system, are between kin and non-kin immediately after eclosion (normal scenario, Unfamiliar treatment), and between kin and non-kin after 24h spent together in equal densities (rarer scenario, Familiar treatment).

We follow the Reviewer’s advice and discuss this now explicitly in the text (lines 234-251 of the “track changes” document).

COMMENT: This is a minor point but Figure 1 (and maybe Figure 2) would look better if they include the data points as well.

Response: Following this advice and the advice of Reviewer 2, we have now replaced Figures 1, 2, 3 and S1 with plots that show the distribution of all data points in beehive plots. For Figures 1, 3 and S1, we have used the visualisation method introduced by Ho *et al.* (2019) Nature Methods, where in addition to beehive plots of raw data we also show a difference axis, where all groups are compared to the NF (familiar, non-related) group, using bootstrapped sampling-error distribution.

Referee: 2

COMMENT: The manuscript RSPB-2019-1137 by Berg *et al.* examines whether the degree of male-male interaction in the seed beetle *C. maculatus* is mediated by relatedness between individual males and/or their familiarity. The role of kinship in sexual selection / sexual conflict is a highly debated topic, and in need of more empirical studies that can weigh in on its

importance and universality. Overall, I thought the experiment was well designed & executed, enjoyed reading this manuscript, but I did have some questions and concerns.

Response: Thank you!

Broader questions:

COMMENT: After having read the passage in the introduction (lines 124-135), the methods (lines 160-1623) and the discussion (lines 342-344, 362-370) I am still not 100% clear whether the authors are arguing that Lymbery et al.'s study is confounded by the presence of Baker's yeast, especially as they don't require eating as adults (lined 349). Surely there are other differences that could also contribute? How do you reconcile these results? Can it be reconciled?

The presence of Baker's yeast most certainly influenced life history of the beetles in Lymbery and Simmons study. Therefore, it is difficult to compare directly the results of our study, and those of Lymbery and Simmons. However, since our population (as most of the laboratory populations) do not normally forage as adults, we believe it is more prudent and ecologically relevant to use the standard conditions when running fitness assays. Of course, there are could be other reasons for the differences between the two studies. For example, Lymbery and Simmons used black-eyed beans as the host species. Our population lives on mung beans. We note that our study is based on two independent experiments and we also note that the effect sizes in Lymbery and Simmons are quite small.

COMMENT: Line 275: Since in block 2 you measured daily female egg production, could you could also have performed a survival analysis on the ALR as well? This would be relevant to your discussion.

Response: Thank you, this is a very good suggestion. We have now performed such an analysis, and it strengthens our conclusion that the effects of relatedness are post-reproductive, since relatedness does not influence ALR in either Unfamiliar or Familiar treatments. The additional analysis is included in the Methods (lines 327-330 in the "track changes" document) and in the new Table 2.

COMMENT: Line 396- end: I felt that your perception error could be better developed. It was quite brief.

Response: We agree that the argument is brief, but we also think that it is difficult to develop it much further without a full-scale research programme to back it up. We do cite another study that mentions this possibility. The main thrust of this project was to evaluate the kin selection hypothesis, and to provide directions for future research in this field.

Minor points:

COMMENT: Line 67: Here, “viscous” should be defined as having genetic structure, otherwise the interpretation is more open.

Response: Done

COMMENT: Line 160-163: This phrasing is odd. Your statement says there is no obvious benefit to yeast, but it is also impossible to tell if there was no benefit to yeast.

Response: We agree. However, this was not our design. As discussed above, Baker’s yeast will most certainly affect behaviour and life history decisions. It is difficult to know how, because this is a novel food source, and having access to food at the adult stage is a novel life history in general. For these reasons, we believe that sticking to the life history that represents the recent evolutionary history of the population is a better choice.

COMMENT: Lines 166. When did the population move to Uppsala?

Response: It was in Uppsala in 2006, more than 100 generations prior to the start of these experiments. Added.

COMMENT: Line 173: ‘meaningful’ instead of ‘great’

Response: Done.

COMMENT: Line 222. I think it is worth pointing out that males were only swapped out once @ day 3 rather than multiply (as was done in some of the fly work), as the experiment lasted at most 10 days

Response: Done

COMMENT: Line 257: Please add in citations for the non-base R packages you used (such as blmeco, popbio, car)

Response: All non-base R packages are now cited in the Methods.

COMMENT: Line 259: Please consider including all your R scripts in the supplementary files along with your archived data.

We now include all R scripts.

COMMENT: Line 266: Is your bobyqa optimizer a function in a package, or is there some other source of this? Please cite.

Response: The *bobyqa* optimiser is part of the *lme4* package. This is now explained in line 312 of the “track changes” document.

COMMENT: Line 288-290. To back up this 'if anything' statement, perhaps an analysis that only uses relatedness? What about an effect size statistic to better describe this difference (if there is one)?

We have now added statistics for parameter estimates in Supplementary table 1, so that the reader can assess the effect sizes. Furthermore, we now show the raw data plots, summary statistics and bootstrapping analyses.

However, we are unsure how an analysis containing only relatedness would be preferable to the current analysis. Since we have a fully factorial 2x2 design, including both factors in the analysis is advisable (as we have done). Furthermore, the new bootstrapping plots and beehive plots provide clear information on the effect sizes.

COMMENT: Line 299-303: While you provide the Wald statistic (z) for the beta coefficients, it would be more beneficial to provide the hazard ratios (& 95% CIs) to give the effect sizes associated with relatedness and/or familiarity?

Response: The in-text statistics are now replaced with a table that also includes the coefficients with their associated standard errors (see Table 2).

COMMENT: Line 313: Kin recognition in fruit flies may also depend on microbiome cues:

Lizé, A., McKay, R., & Lewis, Z. (2014). Kin recognition in *Drosophila*: the importance of ecology and gut microbiota. *The ISME journal*, 8(2), 469.

Lizé, A., McKay, R., & Lewis, Z. (2013). Gut microbiota and kin recognition. *Trends in ecology & evolution*, 28(6), 325-326.

Lewis, Z., Heys, C., Prescott, M., & Lizé, A. (2014). You are what you eat: gut microbiota determines kin recognition in *Drosophila*. *Gut microbes*, 5(4), 541-543.

Response: We agree this is a possibility in *Drosophila* but this was not the prime focus of the current study. Specifically, beetles commonly live in environments where they are exposed to only one plant host and therefore one food type. We included two of these references in the paper as suggested by the Reviewer.

COMMENT: Line 382: Diversity in mates increases the chance of genetic compatibility

Response: Yes, we agree. Nevertheless, there could be other reasons as well, and this was not the focus of the current study.

COMMENT: Figure1 (& S1): Mean +/- SE plots are not as meaningful as boxplots or violin plots as describing your data.

In line with this comment and the similar comment of Reviewer 1, we have now replaced figure 1, 2, 3 and S1 with bee-swarm plots of raw data and, when appropriate, bootstrapped estimation statistics (following Hu *et al.* 2019. Nature Methods). This is probably the best way to visualise raw data and effect sizes.

COMMENT: Finally: Your format of your Raw data as a pdf of an excel sheet is not accessible, and .csv files should be uploaded as ESM.

Response: We have now uploaded our raw data as CSV files instead of PDF files.

Appendix B

Response to Reviewers

Dear Editor,

We were happy to see that both Reviewers and the Associate Editor were very positive about our paper. Reviewer 2 was pleased with how we dealt with original comments and had no further questions or queries. Reviewer 3 had very few minor edits, which we address fully in this revision. Finally, the AE added several important and constructive comments that we also fully incorporated in the revised version.

Below please find our point-by-point response to AE and both Reviewers.

We hope that you will find this version suitable for publication in your journal.

Kind regards,

On behalf of all co-authors,

Elena Berg and Alexei Maklakov

Response to AE:

*This ms is a resubmission the original submission has been handled by a different associate editor. It reports an experiment testing the idea that relatedness and/or familiarity among competing males reduces their harm to females. There is some, although not universal, support for this idea from studies in *Drosophila*, and from a recent paper in the seed beetle *Callosobruchus maculatus*. This paper reports an experiment on the latter species which find no such effect. Thus, while the present paper is not conceptually new, it is important in that its results go, in a way, against the grain. All too often publication bias against negative results creates an exaggerated impression of prevalence of a particular phenomenon.*

Thank you!

The ms has been reviewed by two reviewers. Reviewer 1, who is one of the original reviewers, is completely satisfied with the revisions. Reviewer 2 is likewise positive, but brings up a few minor points related to some rather speculative arguments made by the authors, notably the attribution of the differences with a previous paper to yeast supplement or the speculations as to whether kin selection might operate in the seed beetle or in the bulb mite. The present paper provides no data to bear on these issues; the mention of the bulb mite could be cut completely, and while the yeast supplement difference should be mentioned, there is no basis for attributing the difference between studies to this protocol difference.

We modified this as suggested by the AE. Mention of bulb mite is removed and we do not use yeast supplementation as a potential explanation for the differences between studies.

(1) *In the "familiar" treatment the three competing males were kept together for 24 h before being presented with the female whereas in the "unfamiliar" treatment the males were kept singly until being put in trios and immediately introduced to the female (l. 210-212). Thus it is not only that they are unfamiliar with their competitors; they have no experience of interaction with other males at all. At least in Drosophila, prior experience with rivals affects male courtship and mating behavior (several papers by Brettman). "Unfamiliar" is thus a misnomer for the treatment. If the authors really wanted to study the effect of familiarity, the "unfamiliar" males should have been also kept in trios for 24 h, but the trios should have been reshuffled before introducing the female. This seems to have been done by LyMBERY and SIMMONS, so this is another difference between the studies that should be mentioned. Thus, the text should be revised throughout to avoid conveying the false impression that it is the effect of familiarity that is being tested.*

We revised the text and changed it "Familiar" and "Unfamiliar" to "Group" treatment and "Alone" treatment, respectively, under the umbrella of "Social context". Both scenarios represent ecologically viable situations in this species.

(2) *While it is true that kin selection can be predicted to favor reduced sexual conflict when competing males are related, there are other, more plausible reasons for reduced male-male aggression or female harm when then competitors are brothers or why they are familiar with one another. The original Carazo et al paper has been criticized by several authors (ref 29-31) for making unsubstantiated claims about kin selection. The present ms does acknowledge some of these more parsimonious explanations in the discussion; however, the title, abstract and the introduction imply the experiment reported in this paper is somehow testing kin selection. It is not; whatever the results might have been, the experiments performed would not allow the authors to conclude anything about kin selection. This point is also raised by reviewer 2, who points out that the absence of kin recognition does not imply absence of kin selection; in the same way, differential behavior in the presence of kin versus non-kin would not imply kin selection. Therefore, there is no place for kin selection in the title. The abstract and introduction should be restructured to start with the phenomenon of reduced conflict if males are related and/or familiar, and the potential reasons should be briefly reviewed, including kin selection but without giving it undue weight.*

We modified the text as suggested by the AE. The title now refers to within-group male relatedness. We also modified the Summary and Introduction accordingly.

(3) *Related to the above, among the different alternative explanations, the authors focus in the discussion on the "perception error". This is certainly plausible, but a couple of other explanations deserve some discussion, as they are supported by data from Drosophila, notably that females re-mate more when facing a group of genetically more diverse males (Krupp et al 2008 Current Biol.) or that the degree of male harm to females is largely determined by the most aggressive male of the competing group (ref 31). Do we know anything about this in C. maculatus?*

We agree and we included the citation to Krupp et al. in our Discussion. We felt that the idea that the degree of harm is determined by the most aggressive male is not the most parsimonious explanation.

(4) The ms seems rather long, and the additions suggested above would make it even longer. However, there is a potential for tightening the text. Notably, the introduction gives a detailed chronological story of the different studies who found or did not find the focal effect in Drosophila, and this can be tightened. Similarly, it is quite intuitive why kin selection might reduce male-male aggression and female harm, do that part could also be shortened.

We tightened the manuscript a bit by removing the section about bulb mites and parts of the yeast discussion, and by reducing the discussion of the different studies in the Introduction. The manuscript is now within size limits allowed by Proc B.

Minor points

l. 99. Hollis et al 2015 explicitly set out to test if "familiarity" (or more precisely, rearing brothers together) was necessary for the reduced female harm in the context of the earlier Carazo et al paper. They did not ask if it was sufficient, and thus an unrelated-familiar treatment was not necessary. So referring to the absence of such a treatment as a weakness is being unnecessarily judgmental.

Removed.

l. 317: "between the four treatments" - should be "among"?

Done.

l. 428: The "perception error" due to converged CHC as a potential explanation for reduced female harm when males are kept together was first proposed by Hollis et al 2015, and it would be fair to acknowledge this here.

Done.

Referee: 2

I am very pleased to see that the authors have addressed clearly & completely all of the issues I raised in my first review of the manuscript.

Thank you!

Referee: 3

This paper has undergone round(s) of reviews before I've seen it. Overall I think it is very good shape: solid work, well written and clear manuscript, with a largely balanced treatment of the field. I've only a few of comments to do with interpretation where I think pretty minor edits could potentially improve the ms. However, I

recognise that some of these are relatively subjective: it is up to the authors and editor as to whether these are incorporated.

1) *The life in storage environments, egg laying in clusters, and immediate mating upon eclosion are used to argue that *C. maculatus* are a ripe system for testing the kin selection mediated sexual conflict ideas. But presumably what really matters is viscosity, dispersal, and whether males can encounter socially variable environments in their lifetime (i.e. sometimes brothers, other times not)? I suspect the answer is that we don't know much about these things, just as we don't for *Drosophila* either (or indeed most small insects). A note to that effect I think would balance things up.*

We mention in the Discussion that it is a bit difficult to imagine that fruit flies are likely to encounter brothers (Discussion, paragraph 1). We also mention that in *C. maculatus* such associations are perhaps more likely (Discussion, paragraph 2).

2) *I don't think the authors provided a particularly satisfactory response to Referee 2's first point, about the yeast difference in the Lymbery paper. The text in the main ms seems to be insinuating that there was something weird about the yeast treatment in the Lymbery study which somehow undermines it. It would be nice if this was toned down somewhat. I don't know enough about *C. maculatus* to say either way, but I can't see any obvious reason why one should believe the results of one study over the other. The statement on Line 399-401 "... the presence of live yeast interacts with male behaviour..." seems unjustified. They are different experiments on different beetles in different labs, so there could be loads of reasons for the differences. It is, superficially at least, hard to see how yeast could explain it.*

We removed the statement that yeast could be a sole explanation for potential differences. We retained the discussion of the yeast, and that fact that *C. maculatus* commonly do not forage as adults to provide the general context for the reader.

3) *Line 421-423 argue why bulb mites are particularly suited to the evolution of kin selected benefits, due to rapid population growth and colonisation of new patches. Isn't this true for lots of insects too? I don't obviously see how this argument makes bulb mites especially more suited than beetles or flies.*

We agree and we removed it.

4) *A recognition failure of kin does not automatically mean something non-adaptive. Recognition or not is about mechanisms not cost and benefits. For example, if females of a hypothetical species failed to recognise related brothers as potential mates, and therefore rejected them, we'd probably interpret that as a mechanism for adaptive inbreeding avoidance. I.e. there could be adaptive recognition failure. I do think its perfectly reasonable to make recognition failure a parsimonious mechanism, and also to suggest that its non-adaptive. But it's conceptually useful to articulate that one does not necessitate the other.*

We agree that kin recognition is not necessary for kin selection, and we double-checked the text to make sure that we did not imply that.